# MOMA-LRG: Language-Refined Graphs for Multi-Object Multi-Actor Activity Parsing

**Zelun Luo,     Zane Durante,\*     Linden Li,\*     Wanze Xie,**
**Ruochen Liu,     Emily Jin,     Zhuoyi Huang,     Lun Yu Li,**
**Jiajun Wu,     Juan Carlos Niebles,     Ehsan Adeli,     Li Fei-Fei**
Stanford University
{alanzluo, durante, lindenli, wanzexie, ruochenl, emilyjin, zhuoyih, tinally}@stanford.edu
{jiajunwu, jniebles, eadeli, feifeili}@cs.stanford.edu
Website: https://moma.stanford.edu/,
Toolkit: https://github.com/StanfordVL/moma/
Documentation: https://momaapi.readthedocs.io/

## Abstract

Video-language models (VLMs), large models pre-trained on numerous but noisy video-text pairs from the internet, have revolutionized activity recognition through their remarkable generalization and open-vocabulary capabilities. While complex human activities are often hierarchical and compositional, most existing tasks for evaluating VLMs focus only on high-level video understanding, making it difficult to accurately assess and interpret the ability of VLMs to understand complex and fine-grained human activities. Inspired by the recently proposed MOMA framework, we define **activity graphs** as a single universal representation of human activities that encompasses video understanding at the activity, sub-activity, and atomic action level. We redefine **activity parsing** as the overarching task of activity graph generation, requiring understanding human activities across all three levels. To facilitate the evaluation of models on activity parsing, we introduce **MOMA-LRG** (Multi-Object Multi-Actor Language-Refined Graphs), a large dataset of complex human activities with activity graph annotations that can be readily transformed into natural language sentences. Lastly, we present a model-agnostic and lightweight approach to adapting and evaluating VLMs by incorporating structured knowledge from activity graphs into VLMs, addressing the individual limitations of language and graphical models. We demonstrate strong performance on activity parsing and few-shot video classification, and our framework is intended to foster future research in the joint modeling of videos, graphs, and language.

## 1  Introduction

Computer vision is currently undergoing a paradigm shift from models trained on crowd-labeled data [1, 2] to large-scale base models [3, 4, 5, 6, 7] trained on noisy but readily accessible image-text pairs. Video understanding is no exception, with the rise of Video-Language Models (VLMs) [8, 9, 10, 11, 12, 13, 14] that have shown remarkable generalization capabilities on videos from new domains. When compared to fixed-set video classification [2, 15, 16], VLMs are able to learn and represent a wider variety of concepts and demonstrate superior low-shot abilities on many downstream tasks due to the flexibility and open-vocabulary nature of language. Besides, while annotating videos remains

---

\*These authors contributed equally.

36th Conference on Neural Information Processing Systems (NeurIPS 2022) Track on Datasets and Benchmarks.

one of the most laborious processes in computer vision, VLMs utilize widely and freely available video-text pairs [9, 8, 13, 17] facilitating larger and more diverse pre-training at a lower cost.

Despite improved generalization and scalability, the application of VLMs to human activity recognition faces a number of challenges. The first challenge is adapting existing VLMs for *fine-grained, actor-centric activity recognition*. Essential computer vision applications in healthcare, surveillance, and robotics are often characterized by complex human activities involving many concurrent events between actors and objects. On the other hand, existing VLMs are typically trained on noisy, coarse-grained internet data and evaluated on downstream tasks that are focused on high-level video understanding [18, 10, 12]. It is unclear how VLMs can be effectively adapted to and accurately evaluated for recognizing fine-grained activities. The second challenge is the *lack of a single overarching task* for evaluating VLMs on activity recognition. Human activities are often hierarchical and compositional [19, 20, 21], requiring explicit modeling and evaluation at multiple levels of granularity. Existing downstream tasks used for VLM evaluation, such as activity classification [18], activity segmentation [10, 12], video-text retrieval [13, 12, 8] and VideoQA [10, 13], only provide an incomplete assessment of VLM performance on activity recognition. Lastly, the black-box nature of VLMs makes their predictions difficult to interpret. This hinders the application of VLMs to many risk-averse domains [22, 23, 24], where it is necessary to interpret VLM predictions in a structured and symbolic manner.

To address the aforementioned challenges, this paper first aims to standardize an overarching representation of human activities across varying levels of granularity and provide a unified task for VLMs by generating this representation. Inspired by the recently proposed MOMA [20] framework, we introduce **activity graphs** as dynamic graphs that encompass activities, sub-activities, and atomic actions in a video. Specifically, an activity graph is a representation that simultaneously (1) provides a class label on the activity level, (2) provides a dynamic sub-activity label that contains all temporal boundaries and categories of sub-activities, and (3) captures fine-grained atomic actions in multi-object multi-actor settings with spatial localization and tracking of all entities and temporal localization of all predicates. Further, we introduce **activity parsing** as the overarching task of predicting the activity graph from a video, thereby achieving multi-level activity recognition via activity graph generation.

Next, we introduce the **MOMA-LRG** dataset, a novel activity recognition dataset that leverages both the descriptive capacity of activity graphs and the expressiveness of natural language. MOMA-LRG involves videos with Multiple Objects and Multiple Actors (MOMA) and is designed to enable models to understand a broad set of human activities. To enable few-shot activity recognition with language, MOMA-LRG provides Language-Refined Graph annotations in a format that enables easy conversion from the structured graph representation into natural language sentences.

Lastly, we introduce **GraphVLM** as our framework for evaluating VLMs on activity parsing, consisting of an activity parsing model and a transfer learning paradigm. We first propose an architecture for activity parsing that can be readily adapted for VLMs, featuring a video stream, a text stream, and video tokenizers shared across all three levels of activity. Although fine-tuning is a widely used transfer learning technique, it requires a fixed architecture and clip sampling approach. In GraphVLM, we propose a transfer learning approach based on knowledge distillation, which enables the adaptation of VLMs in a flexible and lightweight manner.

## 2  Related work

**Activity recognition.** Activity recognition tasks a model to identify events performed by human agents. The dominant task has been activity classification on benchmarks such as [2, 15, 16], but other datasets add richer information to their annotations such as spatio-temporal scene graphs [19, 21]. 3D CNNs that jointly model space and time [32, 33, 34] have been popular for activity recognition historically, but recently transformers-based methods have achieved comparable or superior results [35, 36, 37, 38, 39]. Action localization datasets like ActivityNet [15], THUMOS'14 [40], and FineGym [27] label the temporal boundaries and the action class of each action that happens within the video. The methods utilized here tend to propose temporal boundaries and then classify them [41, 42, 43, 44, 45], while others attempt to jointly model both [46, 47]. Other datasets [25, 28] add a spatial dimension, attempting to localize actions in both space and time, where methods include long-term feature banks [48], relation-modeling [49, 50] and pretrained video modules and object

Table 1: A comparison of MOMA-LRG's vocabulary with related video datasets. MOMA-LRG's hierarchy unifies several definitions together (src: source, trg: target, atr: actor, obj: object, c: classified, g: grounded, t: tracked).

| Dataset | Unary predicate | | | Binary predicate | | | | |
|---|---|---|---|---|---|---|---|---|
| | Name | src_atr | src_obj | Name | src_atr | src_obj | trg_atr | trg_obj |
| AVA [25]/AVA-Kinetics [26] | Pose | g,t | - | Person-person/object interaction | g,t | - | - | - |
| Action Genome [19] | - | - | - | Relationship | g | - | - | c,g |
| FineGym [27] | Sub-action | - | - | - | - | - | - | - |
| Home Action Genome [21] | - | - | - | Relationship | g | - | - | c,g |
| MultiSports [28] | Action | g,t | - | Action | - | - | - | - |
| Something V2 [16] | - | - | - | Human-object interaction | - | - | - | c |
| DALY [29] | Action | g,t | c,g | - | - | - | - | - |
| MEVA [30] | Activity | g,t | g,t | Activity | g,t | - | g,t | g,t |
| TITAN [31] | Individual Atomic Actions Vehicle State/Action | c,g,t | c,g,t | Communicative Contextual/Transportive | c,g,t | - | c,g,t | c,g,t |
| MOMA-LRG | Attribute | c,g,t | c,g,t | Relationship | c,g,t | c,g,t | c,g,t | c,g,t |

detection architectures [28]. MOMA-LRG encompasses and extends these existing video datasets by annotating actions, sub-activities that compose them, and rich scene graphs to describe the interactions between entities in a crowded scene.

**Video and language models.** Several works pre-train large-scale models jointly on video and language data for a variety of downstream video-language tasks, such as video captioning, VQA, and video-text retrieval [51, 18, 12, 10, 14, 13]. Pre-training these models often either relies on a combination of masked-language-modeling (MLM) and masked-frame-modeling (MFM) [18, 10, 14] or contrastive learning [51, 9, 12, 13]. These VLMs have shown promising zero-shot results for activity recognition tasks such as activity classification [18], action segmentation [12], and action step localization [12, 10]. These methods show powerful zero-shot capabilities for high-level video understanding, however, they lack explicit knowledge about fine-grained interactions between actors and objects.

**Fine-tuning large pre-trained models.** Finetuning pre-trained large language models for downstream tasks has recently become the most popular learning method in NLP [52, 53]. Methods for efficiently fine-tuning these large language models, such as using adapter modules [54] and prompting [55, 53], use only a small number of learnable parameters while keeping most of the pre-trained model frozen. As a result, these methods are less computationally expensive than full fine-tuning of the entire model, which is the traditional fine-tuning method used in computer vision [56, 57]. There has been recent work adopting these efficient fine-tuning techniques for vision-language models. [58] and [59] both propose adapter module methods and demonstrate comparable performance to full fine-tuning. On the prompt-tuning side, Zhou et al. [60] develops prompt-tuning techniques for vision-language models for zero-shot image classification, Yao et al. [61] uses prompt-tuning for grounding referring expressions in images, and Ju et al. [62] uses image-level tokens and textual prompt tuning for few-shot action recognition.

**Low-shot activity recognition.** Low-shot activity recognition, which recognizes activities that were either scarce or missing from the training set, reduces the reliance on obtaining expensive labels for crowded scenes. Zero-shot action recognition tries to predict unseen classes, where approaches either project visual features into a semantic embedding space [63, 64, 65, 66, 67], an intermediate embedding space learned from textual and visual data [68, 69, 70], or a visual embedding space that is synthesized by incorporating semantic information [71, 72]. Few-shot learning tends to be based on metric learning, which learns similarities to the scarce in-domain training examples [73, 74, 75, 76, 77, 78]. Other recent methods utilized self-supervised and contrastive or meta-learning approaches [79, 80] with a high degree of success. More recently, visual-language models have shown strong results on zero-shot recognition [18, 4, 12].

**Visual grounding and scene graphs.** Visual grounding merges visual and language understanding by attempting to localize an object in an image space given a text query. Some datasets associate nouns with bounding boxes in the video [81, 82], while others introduce scene graph annotations that describe the relationships between entities in the image [83]. Yu et al. [84] and Chen et al. [85] demonstrate that using hierarchical text-generated scene graphs allows for better representation of fine-grained semantics than using raw text alone. Other work has proposed the use of an action graph to generate novel videos [86, 87], defining an object-centric graph with objects as nodes and edges as

Table 2: A comparison of MOMA-LRG with related video datasets. A dash signifies that the annotation does not exist in the dataset, and n/a indicates that the paper did not report a specific number.

| Dataset | Hours | Levels | Activity | | Sub-activity | | Actor | | Atomic action | | | | | |
| | | | | | | | | | Object | | Unary predicate | | Binary predicate | |
| | | | Classes | Instances | Classes | Instances | Classes | Instances | Classes | Instances | Classes | Instances | Classes | Instances |
|---|---|---|---|---|---|---|---|---|---|---|---|---|---|---|
| AVA [25] | 107.5 | 2 | - | - | - | - | 1 | 424K | - | - | 14 | 424K | 66 | 651K |
| AVA-Kinetics [26] | 638.9 | 2 | - | - | - | - | 1 | 310K | - | - | 13 | 633K | 47 | $\sim$800K |
| Action Genome [19] | 82 | 2 | 157 | 10K | - | - | - | - | 35 | 0.4M | - | - | 25 | 1.7M |
| FineGym [27] | 708 | 3 | 10 | 4.9K | 530 | 32.7K | - | - | - | - | - | - | - | - |
| Home Action Genome [21] | 25.4 | 3 | 75 | 1.75K | 453 | 24.6K | 1 | 24.6K | 86 | n/a | - | - | 29 | 583K |
| MultiSports [28] | 18.6 | 2 | 66 | 37.7K | - | - | 1 | 902K | - | - | - | - | - | - |
| Something V2 [16] | 121 | 1 | - | - | - | - | - | - | a few thousand | 30K | - | - | 174 | 318K |
| DALY [29] | 31 | 1 | 10 | 3.6K | - | - | n/a | n/a | n/a | n/a | - | - | - | - |
| MEVA [30] | 9.3K | 1 | 37 | n/a | - | - | n/a | n/a | n/a | n/a | - | - | - | - |
| TITAN [31] | 3 | 1 | - | - | - | - | 3 | 395K | 2 | 249K | 16 | 935K | 28 | 426K |
| MOMA [20] | 66 | 3 | 17 | 373 | 67 | 2364 | 20 | 80K | 120 | 80K | 52 | 12K | 23 | 119K |
| MOMA-LRG | 148 | 3 | 20 | 1.4K | 91 | 15.8K | 26 | 740K | 126 | 396K | 13 | 704K | 52 | 1.4M |

actions with temporal annotations. Wang et al. [88] models videos as space-time region graphs that capture long-range dependencies and spatial-temporal relations between objects.

## 3 Activity Graphs and the MOMA-LRG Dataset

MOMA-LRG improves and extends MOMA [20] by providing the new abstraction of the activity graph as the single universal representation of human activities that encompasses video understanding at the activity, sub-activity, and atomic action levels. Thus, our new formulation for the task of activity parsing as activity graph generation allows for a single overarching task for hierarchical video understanding. The MOMA-LRG dataset also enables the training of few-shot video-language models by encapsulating high-level and fine-grained semantics within activity videos.

### 3.1 Activity Graphs

The key abstraction of MOMA-LRG is the activity graph, an all-encompassing and human-interpretable representation of human activities that captures temporal changes and compositionality. An activity graph is a dynamic graph $\mathbf{G} = [G_1, G_2, \ldots, G_t]$ represented as an ordered list of timed events, such as the addition or deletion of nodes and edges over time. Each $G_i \in \mathbf{G}$ can be represented as the pair $(V_i, E_i)$, where $V_i$ is a set of entities and their attributes and the set of edges $E_i = \{(v_{1i}, r_i, v_{2i}), \ldots\}$ encapsulates the relationships between the source and target entities. An activity graph has two levels of labels: (1) an activity label, which stays constant for the entire graph; (2) a dynamic subactivity label, which changes when subactivities begin and end. Each unique activity instance is associated with a unique activity graph. Unlike the dynamic scene graphs from [19], activity graphs in MOMA-LRG are *activity-centric* and contain information relevant only to the activity. The activity graph includes three levels of hierarchy:

**Activity.** An activity is an event where several human (actors) and non-human (objects) entities interact to complete a multi-step task. Parsing the activity returns an activity class label associated with the activity graph.

**Sub-activity.** A sub-activity is a step that makes up part of a larger activity and is (1) temporally localized within an activity and (2) mutually exclusive between activities. For example, the sub-activity `the adult is comforting the child` is unique to the activity of `babysitting`. Sub-activities are represented with two labels: 1) a temporal boundary, which indicates the start and end time relative to the activity video; 2) a semantic label, which represents the class of the sub-activity. Parsing the activity produces the dynamic sub-activity label that contains the temporal boundaries and class of all sub-activities.

**Atomic action.** An atomic action describes how entities interact within a sub-activity video, which involves understanding entities and their predicates. Atomic actions are **entity-centric**, i.e. entities involved in an atomic action are spatially and temporally localized. Entity labels are entity-centric to disambiguate which entities are involved in a given atomic action. Atomic actions are activity and sub-activity agnostic—that is, a given atomic action class can be involved in many different sub-activity and activity instances. The predicates are atomic, such that they are generic across all activities. Predicates like `running`, `walking`, or `bending` are general and can be involved in

Figure 1: An example of the results for the activity parsing task. For the activity level, the model predicts the activity class `haircut` for the video input. For the subactivity level, the temporally localized sub-activity predictions are displayed on the bottom, with the corresponding sub-activity classes on the legend placed on the left. For the atomic action level, the model has localized and tracked all entities (actors and objects) and predicted their interactions as displayed in the graph visualization on the right. Note: this graphic is a live animation that can be viewed in an Adobe Acrobat PDF viewer.

multiple sub-activity and activity instances. At this level, activity parsing evaluates the ability of the model to predict: (1) all predicates present in the global context, similar to scene graph generation and relationship retrieval, and (2) all the predicate-specified entities across time, similar to spatio-temporal atomic action detection in [25].

Atomic actions consist of two components: entities and predicates. An **entity** is defined to be either a human actor or an object that is present in the scene and relevant to the action being performed. In a video frame, we annotate each entity with a bounding box, class label, and instance ID. Throughout the video instance, an entity is therefore represented as a spatio-temporal tube with a corresponding semantic label. A **predicate** to describe an interaction that occurs with at least one entity. There are two different types of predicates: a unary predicate defined on a specific entity is called an **attribute**, whereas a predicate defined on two or more entities is called a **higher-order relationship**. Unlike other scene graph datasets [83, 19], relationships in MOMA-LRG can involve two or more actors. To do this, we provide **hyperedge** annotations where higher-order interactions involving multiple entities are grouped into a single edge (e.g. multiple actors `beneath` an object). Note that multi-node edges can easily be converted to a set of binary edges if needed.

**Intuition and advantages.** The activity graph is a single universal representation of human activities, consisting of three levels of hierarchy ranging from coarse to fine-grained: activity, sub-activity, and atomic action. This is inspired by the fact that complex human activities in real-world settings are usually hierarchical and compositional across space and time. In particular, complex human activities typically involve a number of achievable steps (activity → sub-activity). It is also essential to understand the roles of actors, the affordances of objects, and the relationships between these components in order to recognize fine-grained activities (sub-activity → atomic action). In contrast, many existing activity recognition benchmarks and tasks [2, 15, 89] only focus on a specific level of granularity.

## 3.2 The MOMA-LRG Dataset

**Dataset statistics.** MOMA-LRG contains 148 hours of videos and provides annotations on 1,412 activity instances from 20 activity classes, ranging from 31s to 600s and with an average duration of 241s. Besides, it contains 15,842 sub-activity instances from 91 sub-activity classes, ranging from 3s to 31s and with an average duration of 9s. On the atomic action level, we provide 161,265 atomic

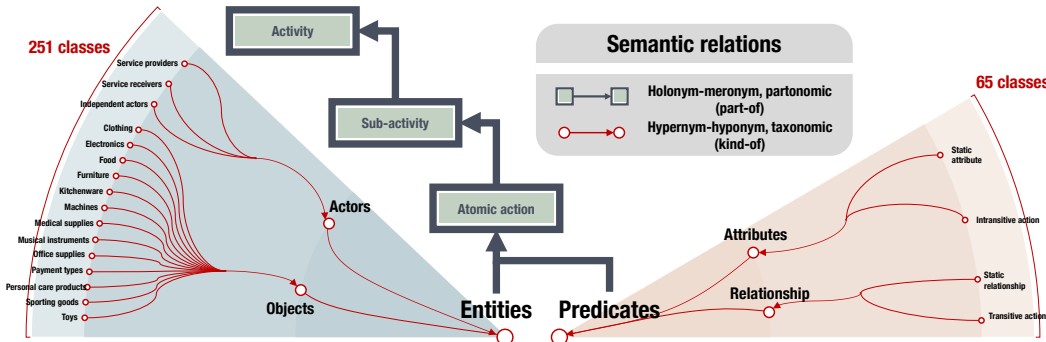

Figure 2: Partonomic and taxonomic hierarchies of MOMA-LRG. MOMA-LRG breaks down activities into sub-activities, which are in turn described by atomic actions. Atomic actions are broken down into entities (actors and objects), whose interactions with each other are described by predicates that either attributes (unary, involving one entity) or relationship (binary, involving two entities).

action interaction instances, which can be further broken down into actors, objects, relationships, and attributes. Specifically, MOMA-LRG contains

- 104,564 actor instances (636,194 bounding boxes) from 26 classes;
- 47,494 object instances (349,034 bounding boxes) from 225 unique classes;
- 1,037,319 relationships from 52 classes;
- 704,230 relationships from 13 classes.

**Language-Refined Graphs.** One of MOMA-LRG's distinguishing features is that it enables few-shot capabilities. To do this, we provide graphical annotations that are easily compatible with natural language through two conventions. First, predicate classes are of the form `[src] [predicate] [trg]`, where `src` is the source entity and `trg` the target entity. This enables easy conversion to natural language given graphical annotations. For example, given an outgoing predicate edge with class `[src] talking to [trg]` from the entity `cashier` onto the entity `customer`, we can produce the sentence `the cashier is talking to a customer`. Second, all of our annotations are in the present continuous tense, e.g. `the player is throwing a frisbee`, which resembles a live narration in a fashion similar to existing video-language datasets (e.g. YouCook2 [90], HowTo100M [9], etc.) created from instructional YouTube videos.

**Comparison with existing datasets.** Compared to existing datasets, there are several key advantages that the MOMA-LRG dataset provides. First, MOMA-LRG grounds all associated entities. In contexts with more than one entity, it is necessary to disambiguate which entities are involved in a particular interaction. Existing ego-centric datasets [16, 27] dodge this issue since at most interaction is involved in a scene. Second, we classify each actor's role. Typical datasets [27, 28] involve one type of actor and hence do not label the person's role [25, 19, 21]. In a diverse set of scenes, the role of the actor becomes more important in understanding actions since it can provide an important signal in parsing a human activity [91]. Third, the MOMA-LRG dataset differentiates between static and dynamic predicates. For example, the dynamic predicate `sitting down` is a dynamic movement where an actor transitions from the `standing` static predicate to the `sitting` static predicate. We argue that observing state transitions is important for the model to learn, encouraging it to learn perceptual causality [92]. For a more detailed comparison, refer to Table 2.

**Comparison with MOMA [20].** First of all, MOMA-LRG introduces a new dataset and a new abstraction of human activity. MOMA-LRG contains an order of magnitude more annotations, along with longer videos from a greater variety of scenes. In addition, MOMA-LRG introduces activity graphs as the overarching graphical representation across all three levels of hierarchy, as opposed to only the atomic level. Secondly, MOMA-LRG is directly motivated by the rise and limitations of VLMs. While VLMs have demonstrated remarkable generalization on videos from new domains and improved scalability through training on free video-language pairs, there is a lack of a single overarching task for evaluating VLMs on complex activity recognition. MOMA-LRG introduces a

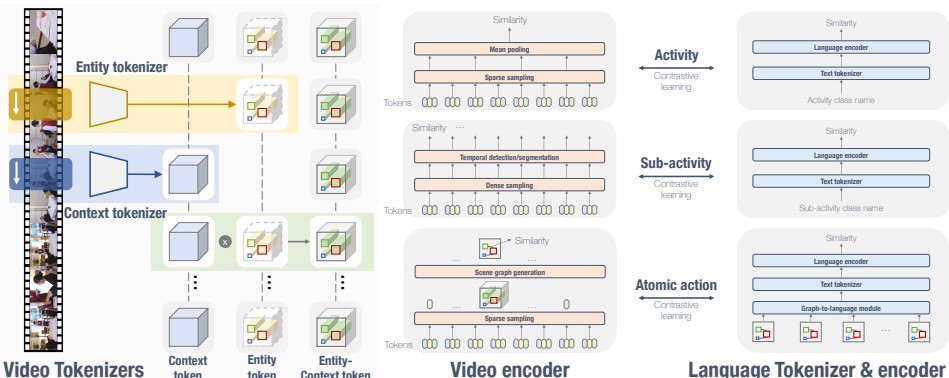

Figure 3: The GraphVLM model: We utilize two different tokenizers for entities and contexts, and a task specific head for each level of the MOMA hierarchy.

new annotation schema that can be easily converted from natural language to graphical annotations to enable few-shot learning, and a new framework (GraphVLM) to evaluate VLMs on activity parsing.

**Ethics.** Prior research [93] shows that Youtube videos exhibit geographic bias. To mitigate potential ethical issues associated with the dataset, we have adopted the following protocols: (1) Taxonomy selection: We carefully selected each activity class in taxonomy to ensure they are gender-neutral, culturally inclusive, and friendly toward people from different socioeconomic backgrounds. (2) Video selection: A team of diverse researchers from different ethnicities and genders selects, examines, and discusses each video to ensure it is diverse and does not contain offensive content. We used keywords from multiple different languages (including but not limited to English, Chinese, French, and Japanese) and word choices to search for videos. Diversification of the videos not only enhances the robustness and generalization of our models, but also significantly reduces the potential bias in the dataset.

## 4 Activity Parsing and the GraphVLM Model

In this section, we introduce a method for performing activity parsing and provide a transfer learning framework to adapt Video-Language Models (VLMs) to activity parsing.

### 4.1 Activity Parsing

We define activity parsing as the task of generating an activity graph as defined in Section 3.1. Specifically, given a video as input, an activity parsing model (1) returns an activity class label, (2) temporally localizes and classifies all sub-activities, as well as (3) localizes each entity in the scene. Following this, it will need to detect all predicates: i.e. all unary predicates (i.e., attributes) involving a single entity and all binary predicates (i.e. relationships) which are between pairs of distinct entities. Refer to Figure 1 for an example of the end results of activity parsing.

### 4.2 GraphVLM: Video Stream

Our video stream module consists of two tokenizers and an encoder for each level of the activity hierarchy.

**Tokenization.** The first tokenizer is the context tokenizer, which consists of a clip sampler and a clip feature extractor. The clip sampler takes a video as input and samples non-overlapping short video clips. It has two parameters: the number of sampled frames $T$ and temporal stride $\tau_c$, meaning that each sampled clip consists of $T \times \tau_c$ total frames. For our clip feature extractor, we evaluate on a Swin-B [36], MViT-B [94], and SlowFast-R50 [32] pre-trained on Kinetics-400 [2]. The second tokenizer is the entity tokenizer, which consists of a frame sampler and an entity detector. The role of the frame sampler is to uniformly sample frames across the whole video, parameterized by $\tau_e$, where $\tau_e$ is the temporal stride (i.e. we sample one frame every $\tau_e$ frames). After frame sampling, we detect entities at the frame level and generate bounding boxes as well as ROI features associated with each

entity, which we call the entity token. These two tokens are used to generate an *entity-context token* by applying the bounding boxes extracted from entity tokenization to the context tokens through ROIAlign.

**Entity detection.** To detect entities and extract entity tokens for activity parsing, we use a Faster-RCNN [89] object detector with a ResNet-101 [95] FPN [96] backbone pre-trained on ImageNet [1]. We use maskrcnn-benchmark[2] for our implementation. For our activity parsing experiment, we treat all human role classes as a single class to facilitate downstream predicate classification. Separately, we also experiment with object detection using the role classes in our dataset. We use Detectron2[3] for actor role detection, and use pre-trained weights from COCO keypoints[97], with the same architecture described above.

**Activity encoding.** The video encoder for activity videos *sparsely* samples $N_a$ context tokens produced by the context tokenizer and performs a mean pool to get the activity feature. This encoding works both with and without a text stream: using the features, we can train an action classifier using a cross-entropy loss exclusively on the features or train jointly with the text stream utilizing a contrastive loss.

**Sub-activity encoding.** The video encoder for sub-activity videos *densely* samples $N_s$ context tokens which are used to run either temporal action detection or segmentation. For temporal action detection, we input the context tokens into a G-tad [41] model and train a model to predict temporal boundaries using a cross-entropy loss. For temporal action segmentation, the encoding is flexible to work with and without the text stream: we can train a classifier with cross-entropy loss using only the features and classify each token as belonging to a sub-activity class or a background class and also train jointly with a text stream using a contrastive loss.

**Atomic action encoding.** Atomic action encoding consists of two parts: per-frame scene graph generation and spatio-temporal atomic action segmentation. For scene graph generation, relationships are grounded over all entities in the scene, whereas spatio-temporal atomic action segmentation is actor-centric and only considers a single actor at a time. We use entity tokens (i.e. object labels, bounding boxes, and ROI features) as input for scene graph detection. We train a RelDN [98] model for scene graph detection using Microsoft's Scene Graph Benchmark[4] for our implementation. We evaluate our model on the tasks of predicate classification, scene graph classification, and scene graph detection without graphical constraints as in Xu et al. [99], since the MOMA-LRG dataset often contains multiple relationships for a given source and target entity. For spatio-temporal atomic action segmentation, the sequence of entity-context tokens of an actor is taken as input, and the model outputs frame-level predicate labels for the actor in a multi-label classification setting. In our implementation, we train a single-layer classifier with a sigmoid activation function, though we note that our framework is compatible with using the generated natural language predicate sentences as supervision via contrastive learning.

### 4.3 GraphVLM: Text Stream

In order to effectively leverage the natural language capabilities of VLMs, we convert all levels of the MOMA-LRG activity graph hierarchy to natural language via our *graph-to-language* module.

**Graph-to-language module.** At the activity level, each class name is a noun, thus it can be represented by its class name or via prompting (e.g. by prepending `"A video of [CLS_NAME]"`). At the sub-activity level, class names are descriptions of the sub-activities in the present continuous tense (narration-style). At the atomic action level, we tag all predicates with `[src]`, and `[trg]` templates to allow for easy conversion into a full grammatically correct sentence in its present continuous form. For example, the predicate `touching` is represented as `[src] touching [trg]`. So, given the entities `[src]=person` and `[trg]=table`, the sentence is `A person is touching the table`.

**Text encoding.** After converting the associated activity graph level to language, we use a pre-trained language model to encode the text. When evaluating existing VLMs (e.g. VideoCLIP [12], FiT [8])

---

[2]github.com/facebookresearch/maskrcnn-benchmark
[3]github.com/facebookresearch/detectron2
[4]github.com/microsoft/scene_graph_benchmark

Table 3: Detection of sub-activities in activity videos with temporal action detection. AP is reported at thresholds 0.1, 0.3, and 0.5 for different backbones.

| Backbones | Pre-train | Temporal Detection | | |
| --- | --- | --- | --- | --- |
| | | AP@0.1 | AP@0.3 | AP@0.5 |
| MVIT-B | K-400 | 17.906 | 9.369 | 5.107 |
| SlowFast-R50 | K-400 | 21.797 | 11.782 | 4.904 |
| Swin-B | K-400 | 22.102 | 10.853 | 4.860 |

Table 4: Activity and sub-activity video classification. Results are reported for activity and sub-activity classification with different video backbones.

| Model | $T \times \tau$ | Pre-train | Activity | | Sub-activity | |
| --- | --- | --- | --- | --- | --- | --- |
| | | | acc@1 | acc@5 | acc@1 | acc@5 |
| MVIT-B | $16 \times 4$ | Kinetics-400 | 0.7731 | 0.9468 | 0.6032 | 0.9473 |
| | $16 \times 4$ | None | 0.5140 | 0.8010 | 0.4375 | 0.7500 |
| SlowFast-R50 | $8 \times 8$ | Kinetics-400 | 0.7569 | 0.9375 | 0.5625 | 0.9226 |
| | $8 \times 8$ | None | 0.4375 | 0.7500 | 0.3739 | 0.7731 |
| Swin-B | $4 \times 3$ | Kinetics-400 | 0.8576 | 0.9688 | 0.6450 | 0.9781 |
| | $4 \times 3$ | None | 0.5282 | 0.8415 | 0.3817 | 0.7868 |
| GCN | $30 \times 1$ | None | 0.7837 | 0.9539 | 0.3829 | 0.8276 |
| GCN (oracle bbox) | $30 \times 1$ | None | 0.9502 | 0.9964 | 0.563 | 0.9706 |

using our framework, we use their respective text encoders. For our model agnostic use-case, we use bert-base-uncased [7].

## 4.4 GraphVLM: Few-shot and Transfer Learning

MOMA-LRG includes a few-shot split, which splits the MOMA dataset into non-overlapping activity and sub-activity classes. The few-shot training set contains 10 activity classes and 45 sub-activity classes, the validation set contains 5 activity classes and 24 sub-activity classes, and the test set contains 5 activity classes and 22 sub-activity classes. For our baseline methods, we report results using two meta-learning classifiers, OTAM [100] and CMN [101]. To evaluate the performance of video-language models in the few shot setting, we perform out-of-the-box classification for a pre-trained VideoCLIP [12] and Frozen-In-Time [8] video-language model on activity and sub-activity classification on the meta-test set. We use class names as text (either activity or sub-activity) and raw videos as input. To compute the class label for a video input, we find the text embedding that is closest in dot product similarity to the video embedding as in Xu et al. [12]. We visualize the performance of this method on the regular MOMA-LRG test set in Figure 4. For $k$-shot video classification, we sample $k$ videos per class and average the representation to obtain a prototype video. We compute a weighted average between the text embedding and the video prototype and classify using the same method as in the zero-shot setting. Details for our explanation and an ablation study of our method can be found in the Appendix.

In addition to our framework for evaluating VLMs without training, we also propose a method for using VLMs for activity parsing in a more flexible manner than full fine-tuning. We use knowledge distillation to incorporate visual and linguistic knowledge from VLMs into the activity parsing framework. This is considerably more flexible than full fine-tuning since it is model-agnostic. In this method, we can use a different backbone network than that used by the VLMs, and can sample clips differently so long as the clips from the student and the teacher model are centered at the same frame. We report results using our framework and investigate the effect of incorporating linguistic information for spatio-temporal atomic action segmentation in Table 5. Details for our approach are explained in the Appendix.

## 5 Activity Parsing Evaluation

In this section, we evaluate our dataset on methods across two different tasks. First, we evaluate model performance on the activity parsing task which leverages the hierarchy of our dataset. Next, we examine our method on activity parsing in the few-shot setting.

We evaluate each level of activity parsing using the following metrics.

**Activity**: The performance of activity recognition is measured by the top 1 accuracy (acc@1) and top 5 accuracy (acc@5) for video-level activity classification. Results are shown in Table 4 for the standard setting and Table 6 for the few-shot setting.

**Sub-activity**: The performance of sub-activity recognition is measured on two tasks. First, we report the sub-activity acc@1 and acc@5, where the pre-segmented sub-activity video is used as input. Results are shown in Table 4 for the standard setting and Table 6 for the few-shot setting. Second, we evaluate temporal detection using mAP at thresholds 0.1, 0.3, and 0.5 and report the average mAP following [40]. Results are shown in Table 3.

Table 5: Scene graph detection, entity detection, and spatio-temporal atomic action segmentation results from the methods described in Section 4.2. The entity detection results pose challenges for scene graph detection, as is evidenced by the relatively higher scores for SGCls and PredCls, where ground truth bounding boxes and class labels are known.

| | Scene Graph Detection | | | | Entity Detection | | | | | | | Spatial-Temporal Segmentation | |
| | Recall@20 | Recall@50 | Recall@100 | | AP | AP50 | AP75 | APs | APm | APl | | Video only | Video + text stream |
|---|---|---|---|---|---|---|---|---|---|---|---|---|---|
| PredCls | 58.2243 | 62.4389 | 64.0983 | Actor role | 38.3567 | 58.1256 | 41.2369 | 7.8053 | 19.4897 | 40.0392 | MViT-B | 0.2130 | 0.2353 |
| SGCls | 44.3065 | 48.3825 | 50.2992 | Entity | 15.2896 | 29.3032 | 14.0742 | 4.3134 | 10.1288 | 17.1472 | SlowFast-R50 | 0.1975 | 0.2023 |
| SGDet | 37.6275 | 43.8594 | 47.9960 | | | | | | | | | | |

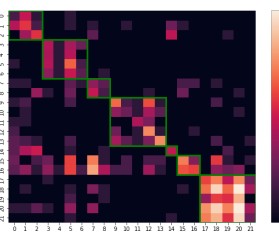

Figure 4: A confusion matrix for zero-shot classification of Video-CLIP [12] on the standard MOMA-LRG sub-activity test set. The sub-activities are ordered to be adjacent to other sub-activities within the same activity. As is indicated by the green squares and the results in Table 6, there is a significant degree of within-activity confusion for zero-shot video language models.

Table 6: Low-shot video classfication. We evaluate both VideoCLIP [12] and Frozen-in-Time [8] within our few-shot framework for activity and sub-activity classification. We note that although the video-language models we tested performed well for high-level activity classification, they performed significantly worse for the more granular task of sub-activity classification.

| | Activity | | | Sub-activity | | |
| Model | 0-shot | 1-shot | 5-shot | 0-shot | 1-shot | 5-shot |
|---|---|---|---|---|---|---|
| OTAM [100] | - | 80.71 | 92.07 | - | **57.14** | **72.59** |
| CMN [101] | - | 73.57 | 86.30 | - | 52.30 | 66.60 |
| VideoCLIP [12] | 75.90 | 84.40 | 84.80 | **30.80** | 32.70 | 32.70 |
| Frozen [8] | **90.80** | **92.30** | **92.50** | 19.10 | 26.50 | 26.30 |

**Atomic action**: The performance of atomic action recognition is measured on two tasks. First, we evaluate entity detection using standard average prevision (AP) metrics. To evaluate scene graph generation, we follow work in [83, 19, 99] and perform the following tasks: predicate classification (PredCls), which takes ground truth bounding boxes and object categories as input and returns predicate labels, scene graph classification (SGCls) which only takes in ground truth bounding boxes as input and predicts object categories and predicate labels, and scene graph detection (SGDet) which simply takes in an input image and predicts bounding box locations, object categories, and predicate labels. Results are shown in Table 5.

## 6 Conclusion

We introduce the MOMA-LRG dataset, a large activity recognition dataset of complex human activities that enables the evaluation and fine-tuning of large, generalizable video-language models. We define activity parsing as the overarching task of activity graph generation, requiring video understanding at multiple levels of granularity. We demonstrate the capacity of MOMA-LRG to train video-language models by introducing a model-agnostic and lightweight approach for adaptation, and we evaluate VLMs by demonstrating strong few-shot classification performance. We hope that MOMA-LRG will enable further research into generalizable activity recognition models that are trained with multiple input modalities or generate language descriptions for videos.

## Acknowledgement

This work was partially supported by the Schmidt Futures, Toyota Research Institute (TRI), the Stanford Institute for Human-Centered AI (HAI), Panasonic, ONR MURI N00014-21-1-2801, ONR MURI N00014-22-1-2740, National Science Foundation grant 2026498, and Bosch, Salesforce, and Samsung. Last but not least, we would like to express our sincere gratitude to Shyamal Buch and Yuliang Zou for their discussions and constructive feedback.

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
