# OpenReview forum: "MOMA-LRG: Language-Refined Graphs for Multi-Object Multi-Actor Activity Parsing"
_NeurIPS.cc/2022/Track/Datasets_and_Benchmarks — NeurIPS 2022 Datasets and Benchmarks _

### Official Review · Reviewer_d2UU · 2022-07-25
**Review of MOMA-LRG: Interesting contribution without accessible dataset and code**

**Rating:** 6
**Confidence:** 4

**Strengths:**

The paper is quite well organized, the topic is current.
The proposed approach via activity graphs to classify activities is very interesting and reasonable approach. Furthermore, the proposed dataset that should provide different levels of granularity of the actions is also meaningful.


**Weaknesses:**

In general I like very much the idea of the paper and the described problem and the proposed solution. Unfortunatelly, I see many shortcomings in the current version.
1. It is not clearly described how the activity graphs corresponding to the videos are created and how they can be used for evaluation.
2. My main concern is about the provided code/dataset URL.
Dataset MOMA-LRG: I tried to explore the dataset and the proposed activity graphs and GraphVLM on the supporting GitHub. Unfortunatelly the documentation is very poor. I managed to install the momaapi (please, put requirements to requirements.txt, also the scripts have different names than they actually have - e.g. visualise -> run_visualise.py). However, there is no description about how to get the MOMA-LRG dataset. The only link I found links to the former MOMA dataset (submission to Nips 2021). Therefore it is also impossible to evaluate the difference of the newly proposed dataset to the former MOMA dataset. It would be sufficient to get some sample of the dataset to explore it.
3. There is also no code for the evaluation of the dataset on different models including the proposed activity parsing and GraphVLM model, or it is not documented how can one make it running. (In appendix is written that it will be available on the GitHub page).
4. Connection to MOMA dataset - in the paper it is mentioned that the dataset differs from the previous MOMA dataset. Mainly by providing new abstractions of the activity graph. After seeing the GitHub page, I am a bit confused about the differences. Is the MOMA-LRG using MOMA dataset and only providing additional abstractions, or is it a new dataset? Both is fine, but the relationship between the dataset should be clear right from the beginning of the paper and instead of presenting a new dataset, presenting maybe "an extension of the MOMA dataset".
5. Some parts of the methods and their connection are not clearly described. The authors should keep in mind that the presented work should serve as a Dataset/Benchmark and as such also clearly present it to the potential user. I see that the activity graphs are the proposed abstraction on top of the MOMA dataset, and they enable evaluation on different level of granularity. Is that right? How is GraphVLM connected to this? In the contributions it is said that the "GraphVLM is framework for evaluating VLMs on activity parsing", but from the description, I don't see how can I use it for evaluating some new VLM method.

Overall, as the submission is for "Datasets and Benchmarks", I think the usability and accessibility of the presented dataset and its usability for benchmarking is crucial. In the current version it is impossible for the user to get to the dataset and use it for benchmarking, see how to evaluate it, as well as to see the results of other methods. Furthermore, without having the access to the proposed dataset and the code for other contributions, the usefulness and novelty of the presented work is hard to evaluate.

Other issues with the GitHub page:
- title is "MOMA" dataset, but in the paper you present MOMA-LRG dataset - I feel a bit confused
- mention the size needed for installation (and for the dataset).
- please, put requirements to requirements.txt
- the scripts have different names in documentation than they actually have - e.g. visualise -> run_visualise.py

Typos/Mistakes
- line 284 "we we report"
- line 216 Figure ??
- line 156 "within in"
- line 125, 126 - "While Wang et al...." - the sentence doesn't sound correct - unfinished?

**Additional Feedback:**

I hope that authors will improve the submission so it can be accepted and actually useful for the potential users. I provide suggestions for improvements and questions in the sections above.

[UPDATE: I raised my score but still have concerns about the connection to MOMA dataset, especially thanks to the confusing Documentation.]

**Clarity:**

The first part of the paper is well written, but I have some issues with interconnection of individual contributions, as mentioned in "Weaknesses" section above.

Some other unclear things:

How are the graphs evaluated for correctness and compared to the ground truth annotations?
How was the ground truth annotation created - how can be the authors sure that it is the only correct or the most meaningful annotation for the given video? How were the produced activity graphs validated?
How the authors can make sure that the NL sentences produced from these "ground truth" activity graphs are then correctly describing the scene? (line 274) - do they somehow evaluate their correctness?
How were the videos/classes selected? Are the videos similar or selected to cover different areas? What type of environment? (e.g. multiple actors, multiple distractors, outside, household, only human actors, animal/object actors such as "car is moving towards a lamp", "dog is barking to the bird"?)
On line 197 is written "MOMA-LRG ground all associated entities" - how is the grounding of all associated entities done and how it can be used for evaluation?

I understand that there is not enough space in the paper itself to explain all the details, but I found also the Appendix material very brief.


**Correctness:**

The dataset structure as described in the paper and the appendix is fine. However some features of the dataset as well as the described evaluations of the dataset are in the current version impossible to evaluate.

**Documentation:**

The documentation is very brief, incomplete and has some issues as mentioned in the Weaknesses above.
GraphVLM has no code or documentation available.
the code itself is not much commented.
Some links on the GitHub page ("Annotations" section) do not work.
Not clear how to get and interact with the dataset.
In checklist - DOI is recomended, or should be discussed, but authors didn't discuss on the topic.

**Ethics:**

It is not clear to me, how were the data for the dataset selected (source is YouTube) so there might actually be some issues with the data or how they got the consent? Is it under CC license? Could authors discuss on the topic?

**Relation To Prior Work:**

The related work section is covering well the recent work in the area. I have the following comments:
1) The relationship to the MOMA dataset is not clear in the submission and should be clarified.
2) In the sections "Fine-tuning large pre-trained models", "Low-shot activity recognition", and "Visual grounding and scene graphs" it is not mentioned how the prior works are connected to the presented contribution. Why the first two sections are actually mentioned in the contribution? I think the authors would do better if they would make the contribution clearer and more focused.

**Summary And Contributions:**

The paper proposes a dataset of complex human activities that are annotated with activity graphs. These activity graphs can be also transformed into natural language sentences. The main contributions of the paper are:
1) MOMA-LRG dataset, an activity recognition dataset with multiple objects and multiple actors
2) activity graphs - dyanmic graphs that can encompass activities, sub-activites, and atomic action in a video and can provide a label on different levels.
3) GraphVLM - framework that can evaluate VLMs on activity parsing

---

### Official Review · Reviewer_MPjz · 2022-07-26
**Review of data track paper 56**

**Rating:** 7
**Confidence:** 4
**Correctness:** I have not checked all the details. T…
**Clarity:** The paper is well written.

**Strengths:**

1. The newly defined activity graphs and the evaluation for the human activity understanding are novel.

2. The proposed model GraphVLM serves as a good framework for evaluating VLMs on activity parsing.

**Weaknesses:**

1. The paper has not explained why the defined activity graphs are a good way to evaluate the ability of the model, or the intuition of proposing it.

**Additional Feedback:**

N/A

**Documentation:**

Yes, detailed documentation has been provided.

**Ethics:**

There is no ethical concern.

**Relation To Prior Work:**

Yes

**Summary And Contributions:**

This paper proposes a dataset for evaluating activity recognition in video. Different from previous datasets that only focus on high-level understanding, it defines activity parsing as the overarching task of activity graph generation and requires the model to have a comprehensive understanding of human activities. In addition, it also provides a VLM by incorporating structured knowledge from activity graphs for this task.

---

### Official Review · Reviewer_tDLt · 2022-07-26
**Review of MOMA-LRG**

**Rating:** 6
**Confidence:** 4
**Correctness:** Yes.

**Strengths:**

+ The MOMA-LRG dataset is a large-scale dataset with complex human activities and detailed activity graph annotations that
can be readily transformed into natural language sentences.  I imagine a lot of careful planning, thought, money, and effort would have been needed to put together the dataset.  It has great potential to drive progress in video-language based human activity understanding.

+ In contrast to how Video-Language Models have been largely evaluated thus far, which is on coarse-grained human activity recognition, the proposed work can facilitate evaluating such models on fine-grained activities at multiple levels of granularity.  This would enable a more detailed way to compare the performance of such models, and also lead to the development of models that can more readily be used for applications involving complex human activities such as those in healthcare and robotics.



**Weaknesses:**

- Although the overall dataset contribution is very good, the evaluation and analysis sections are quite weak and incomplete in comparison.  Specifically, Section 5 on Activity Parsing Evaluation seemed to lack any analysis.  For the results in Table 2-4, what are the takeaway messages?  Section 4.4 describes a method that uses knowledge distillation, but it was not clear where this method was being evaluated and analyzed.

**Additional Feedback:**

My current rating is borderline accept.  I find the dataset to be compelling and of high-value; however, there are weaknesses with the writing and analysis.

**Clarity:**

The introduction, related work, and Activity Graphs and the MOMA-LRG Dataset (Section 3) are clear.  However, I found Sections 4 and 5 to be unclear at times, as pointed out in the weaknesses above.

**Documentation:**

Overall, the main paper and supplementary contain sufficient information.

**Ethics:**

Since the dataset focuses on humans performing activities, it would be helpful if the authors could answer the following question in the ethics guidelines: "Consider whether the data might encode, contain, or potentially exacerbate bias against people of a certain gender, race, sexuality, or who have other protected characteristics. For instance, does the dataset represent the diversity of the community where the approach is intended to be deployed?"

**Relation To Prior Work:**

Yes.

**Summary And Contributions:**

The paper proposes the MOMA-LRG dataset, a large activity recognition dataset of complex human activities for evaluating and fine-tuning large video-language models. The paper defines activity parsing as the overarching task of activity graph generation, requiring video understanding at multiple levels of granularity (activity, sub-activity, atomic actions). The paper also demonstrates the capacity of MOMA-LRG for training video-language models by introducing a model-agnostic and lightweight approach for adaptation, and by evaluating video-language models on few-shot classification performance.

---

### Official Review · Reviewer_tN94 · 2022-07-28
**Great contribution towards incorporating video-language models with multi-grained complex activity recognition tasks**

**Rating:** 8
**Confidence:** 4
**Clarity:** Yes.

**Strengths:**

1. This paper generally inherits the strengths of MOMA. The multi-level granularity and activity-centric properties of activity graph are technically sound. The dataset is punctiliously annotated and the annotations can be converted to natural language sentences.
2. The proposed knowledge distillation based transfer learning is a simple and efficient evaluation method for VLMs.
3. They conduct solid experiments on activity parsing and few-shot learning on the introduced dataset across several VLMs, providing insights to future works.

**Weaknesses:**

Though this paper extends MOMA with an emphasis on video-language models, the most related parts to natural language are graph-to-language module and knowledge distillation from pretrained VLMs for classification tasks. Also the natural language part in this paper is in a relatively limited format. It can be expected to explore other applications like generating language descriptions for videos with the advantage of activity graphs in future works.

**Additional Feedback:**

No.

**Correctness:**

It is constructed in a sound way. The evaluation methods and experiment design appropriate and performed correctly.

**Documentation:**

Yes.

**Ethics:**

No.

**Relation To Prior Work:**

Yes.

**Summary And Contributions:**

Inspired by the MOMA dataset and the activity parsing task in prior works, this paper moves one step further to incorporate video-language models with activity parsing. It proposes 1) activity graphs as a unified representation involving 3 levels of human activities, 2) activity parsing as a task to generate activity graphs and understand complex activities, 3) MOMA-LRG as a large dataset with natural language sentence and activity graph annotations, and 4) a model-agnostic approach for adapting and evaluating VLMs on the dataset.

---

### Review · Ethics_Reviewer_LoPY · 2022-08-25

**Recommendation:** 2

**Ethics Review:**

The technical reviews raised multiple points that have failed to be sufficiently addressed in the current draft of the paper. Specifically, the paper fails to acknowledge [prior research](https://dl.acm.org/doi/pdf/10.1145/2187836.2187870?casa_token=dip47JpO7iUAAAAA:UYjjQnSyWTbvsnmoImxRv-oBK8wnUl-XDRbtXP6xgQG85xfx-PU8xyNQk5tkK04Ahc9qJea6q5Q6Lg)  on the predominate proportion of youtube videos are generated in North America and Europe. The implication of geographic skew is that by sampling a cross section of public videos from the youtube platform, one can unintentionally embed social biases into their dataset. The authors should present some greater details on how the final sample of videos were selected and how they ensured diversity in the  composition of their dataset.

The authors should also address consent and access issues in the final version of the paper. Specifically, the authors state in the paper "Instead of providing the raw videos, we provide YouTube IDs and a script for downloading them. Note that this is a standard practice for many video datasets collected from the Internet, such as ActivityNet [15] and Kinetics [2].  Since we only provide YouTube IDs, only publicly available videos are available for download, and **video owners may disable accessibility at any time.**" However, in the supplemental material the authors state, "The videos in our dataset were collected from Youtube. We provide the dataset annotations and a script to crawl videos below. Alternatively, **we host the dataset on Google Drive for easy access who are interested in the raw videos for research purposes.**" These policies are in conflict and effectively remove the ability of video owners to remove access. This conflict should be reconciled in the final version of the paper.

Finally, while the authors noted in the paper that videos were assess for offensive material, they did not note whether the videos had been screened for copyrighted material. This would be relevant if the authors seek to make dataset available via an open source license (as noted in the supplemental materials).

---

### Meta-Review · Area_Chair_RtcB · 2022-09-10

**Recommendation:** Accept
**Confidence:** 4

**Metareview:**

The work is of a high quality but the reviewers have struggled to understand certain aspects of the documentation and source code in the first version. The authors have worked to provide an improved version but unfortunately only one reviewer engaged in re-evaluating the work.

However, there's always been the question of whether activity hierarchies are essential/helpful for video understanding, so I believe this benchmark will be of value to the community.

The authors are encouraged to keep their documentation up to date and engage with users to facilitate their understanding of annotations and APIs.

I believe the paper is of value to the research community, as a dataset and benchmark and hence recommending acceptance.

---

### Decision · Program_Chairs · 2022-09-16

Accept